# Betulinic Acid-Nitrogen Heterocyclic Derivatives: Design, Synthesis, and Antitumor Evaluation *in Vitro*

**DOI:** 10.3390/molecules25040948

**Published:** 2020-02-20

**Authors:** Yuqin Yang, Tianxin Xie, Xuehao Tian, Nana Han, Xiaojing Liu, Hongshan Chen, Jinchai Qi, Feng Gao, Wen Li, Qianwen Wu, Su Huo, Yuhao Gu, Ziqi Dai, Penglong Wang, Haimin Lei

**Affiliations:** School of Chinese Pharmacy, Beijing University of Chinese Medicine, Beijing 102488, China; 20170941161@bucm.edu.cn (Y.Y.); lw19991103@163.com (W.L.);

**Keywords:** betulinic acid, **BA**-nitrogen heterocyclic derivatives, antitumor, Hela, flow cytometry

## Abstract

Betulinic acid (**BA**) is a star member of the pentacyclic triterpenoid family, which exhibits great prospects for antitumor drug development. In an attempt to develop novel antitumor candidates, 21 **BA**-nitrogen heterocyclic derivatives were synthetized, in addition to four intermediates, 23 of which were first reported. Moreover, they were screened for in-vitro cytotoxicity against four tumor cell lines (Hela, HepG-2, BGC-823 and SK-SY5Y) by a standard methylthiazol tetrazolium (MTT) assay. The majority of these derivatives showed much stronger cytotoxic activity than **BA**. Remarkably, the most potent compound **7e** (the half maximal inhibitory concentration (IC_50_) of which was 2.05 ± 0.66 μM) was 12-fold more toxic in vitro than **BA**-treated Hela. Furthermore, multiple fluorescent staining techniques and flow cytometry collectively revealed that compound **7e** could induce the early apoptosis of Hela cells. Structure–activity relationships were also briefly discussed. The present study highlighted the importance of introducing nitrogen heterocyclic rings into betulinic acid in the discovery and development of novel antitumor agents.

## 1. Introduction

Natural products play a major role in the antitumor drug discovery. Over 60% of antitumor drugs are developed from natural products [1]. Pentacyclic triterpenoids are a class of pharmacologically active and structurally rich natural products with privileged motifs for further modifications and structure–activity relationship analyses [2,3,4,5]. As a lupane-type pentacyclic triterpenoid, betulinic acid (3*β*-hydroxy-lup-20(29)-en-28-oic acid, **BA**, Figure 1) is widespread in many plants. It had been demonstrated that **BA** possessed various bioactivities, including antitumor, anti-HIV, anti-inflammatory, antiviral and antiseptic activities [6,7,8,9,10,11]. Since minimal toxicity against normal cells and antiproliferative activity against a panel of tumors [12], it was recognized as the leading compound of antitumor agents. Moreover, **BA**’s continuous structural modification had been an extremely attractive hot topic worldwide. It consisted of a 30-carbon skeleton which could be modified at three positions, the secondary hydroxyl group (C-3), the hydroxyl group (C-28) and at the alkene moiety (C-20), respectively. It was reported that C-28 carboxylic acid was essential for the cytotoxicity [9,13]. For example, 20, 29-dihydro betulinic acid derivatives were synthesized with IC_50_ less than 0.4 μg/mL [14]. **BA** derivatives modified at the C-3 position [4-nitrobenzyl-oximino] had shown IC_50_ values 0.4 μg/mL against the U-937 cells.

In the last few years, nitrogen-containing heterocyclic derivatives had been synthesized as antitumor agents. For example, the incorporation of an imidazole scaffold at the C-28 or C-3 position of betulinic acid with ester or amide bonds could improve toxic activity significantly; and the majority of the novel compounds were particularly effective against the hepatoma HepG-2 (IC_50_ = 0.8, 1.7, 2.0 μM, respectively) cell line [15]. Eignerova Barbara [16] acetylated the 3 hydroxyl group of betulinic acid and piperidine, the introduced carbon chain connection on the 28 carboxyl group of which showed high and selective cytotoxicity (1.6 mM on G-361 cells). Other N-heterocyclic derivatives [17,18,19] had been reported to possess antiproliferative effects against tumor cell lines.

In the present study, a series of novel **BA**-nitrogen heterocyclic derivatives were designed and synthesized to introduce different nitrogen heterocycles into the 3, 28-hydroxyl of **BA** with the ester condensation reaction. Representative tumor cell lines were applied to evaluate the antitumor activities of these compounds. Cell morphology changes on Hela induced by compound **7e** were observed by 4′,6-diamidino-2-phenylindole (DAPI) staining. Furthermore, fluorescence staining observations and flow cytometric analyses were performed to investigate the potential mechanism.

## 2. Results

### 2.1. Chemical Synthesis

The syntheses of 21 **BA**-nitrogen heterocyclic derivatives were shown in Scheme 1. **BA** was treated with potassium carbonate solution and benzyl bromide/1, 2-dibromoethane in dimethylformamide (DMF) at 85 °C for 4 h to obtain compound **2** and **6**. Then compound **2** was treated with succinic anhydride and chloroacetic acid in DCM at 80 °C for 5 h catalyzed by 1-Ethyl-3-(3-dimethylaminopropyl)carbodiimide/4-dimethylaminopyridine (CH_3_)_2_NC_5_H_4_N) (EDCI)/DMAP), and compounds 3 and 10 were obtained. By further substitution with nitrogen heterocyclic ring (R) or reduction reaction, we got the compounds **4a**–**4d**, **5a**–**5d**, **7a**–**7e** and **11a**–**11e** (Table 1). Compounds **9a**–**9b** (Table 1) were obtained by an oxidation and substitution reaction, starting from compound 6. All **BA**-nitrogen heterocyclic derivatives were determined by ^1^H-NMR, ^13^C-NMR and HR-MS.

### 2.2. Cytotoxicity

The in-vitro cytotoxicity of the **BA**-nitrogen heterocyclic derivatives was evaluated on four pathologic live cells (Hela, HepG-2, BGC-823 and SK-SY5Y) by MTT assays. As shown in Table 2, the IC_50_ of the derivatives exhibited better inhibitory activities against Hela, HepG-2, BGC-823 and SK-SY5Y compared to **BA**. In particular, compounds **7a**, **7e** and **11e** showed stronger inhibitory effects against the four tumor cell lines than the rest of the compounds (Figure 2).

In addition, it was observed that after introducing the nitrogen heterocycle into the 3-hydroxyl or 28-carboxyl of **BA**, it relatively improved their cytotoxicity. As shown in Table 2, the most promising was compound **7e**, which showed higher cytotoxicity than **BA**. The IC_50_ of derivatives compound **7e** were 2.05 ± 0.66 μM, 2.79 ± 0.53 μM, 3.52 ± 0.37 μM and 3.13 ± 0.84 μM against Hela, HepG-2, BGC-823 and SK-SY5Y, respectively. It was further verified that the small molecule nitrogen heterocycle could enhance **BA**’s bioactivity, which was in line with our previous report [20].

### 2.3. Cluster Analysis- Orthogonal Partial Least Squares Discriminant Analysis (OPLS-DA)

To further explore the structure–activity relationship, OPLS-DA was performed for all designed **BA** derivatives. Analyses revealed an antitumor activity discrimination between the different **BA** derivatives. As for the effect of the structure modification site and different nitrogen heterocyclic rings of **BA**, they were divided into two groups according to the difference in the in-vitro antitumor activity (Figure 3 and Figure 4). Through data analysis, we found that the structure modification site on **BA** showed a certain degree of regularity in their effect upon activity, the structural modification at positions C-3 and C-28 could improve antitumor biological activity in vitro, while the structural transformation of C-28 might have more potential to enhance cytotoxicity on the same series of tumor cells; for example, the antitumor activities of compounds **7a**, **7c**, **7d** and **7e** were stronger than compounds **5a**, **5b**, **5c** and **5d**. In addition, different nitrogen heterocyclic rings on **BA** also affected their activity (compound **11e** > compound **11a**, **11b**, **11c** and **11d**). The cluster analysis of OPLS-DA might provide us with further directions for the further analysis of **BA** derivatives. All data were analyzed using SIMACA 13.0. Analysis showed no samples being outside the Hotelling T2 95% confidence ellipse that could influence the analyses, and high values of explained variation and predictive ability were obtained (Table 3). Besides, the values of explained variation and predictive ability were 0.883 and 0.978, respectively, according to OPLS-DA for the IC_50_ of four tumor cells shown in Figure 4. 

### 2.4. Morphological Analysis

To characterize the effects of apoptosis induced by compound **7e** on Hela, the nuclear morphological changes were observed with DAPI staining. After treating with compound **7e** for 48 h, it can be seen from the results that the number of Hela cells was decreased sharply, and the cell space became larger significantly (Figure 5I); moreover, Hela cells showed nuclear morphological changes typical of apoptosis. As pictured in Figure 5II, in the control group, it appeared to have normal cellular morphology, the nucleus was intact, and the cells did not show the characteristics of apoptosis. When treated with **BA**, the number of cells was decreased, the contours of some cells became irregular, nuclear fragmentation was appeared, whereas compound **7e** treatment caused a significant decrease in the number of cells, evident nuclear fragmentation, and did not see an intact nucleus. Thus, the results indicated that compound **7e** could induce apoptosis in Hela cells.

### 2.5. Apoptosis Analysis Using Annexin V-FITC/Propidium Iodide (PI) Staining

To evaluate the apoptosis induced by compound **7e** and to further determine early apoptosis and secondary necrosis, apoptotic rates were analyzed by flow cytometry using an Annexin V-FITC/PI staining. As shown in Figure 6, when treated with different concentrations of compound **7e**, the percentages (Q2 + Q4) of apoptotic Hela increased from 12.6% in control cells to 14.2%, 29.5% and 73.3%, respectively. Furthermore, the results indicated that compound **7e** could induce Hela cells’ early apoptosis in a concentration-dependent manner. It speculated that compound **7e** could induce Hela cells early apoptosis to an antitumor effect.

## 3. Discussion

**BA** is widespread in natural plant and Chinese herbal medicine, used for the prevention and treatment of tumors, and there are large number of betulinic acid derivatives that have been synthesized [21,22,23,24]. In this report, a series of different **BA**-nitrogen heterocyclic derivatives were designed and synthesized to improve their biological activity and hydrophilicity. After introducing a different nitrogen heterocycle in the 3-hydroxyl/28- carboxyl of **BA** using the ester condensation reaction, the majority of these derivatives showed much stronger cytotoxic activity than **BA**.

In chemical synthesis, introducing succinic anhydride in the C-3 of **BA** was explored, and the reaction solvent was changed from THF to DCM. This reaction was simple, mild and controllable, with a yield of 79%, which was suitable for the synthesis of such compounds in the future. In the structure–activity relationship, we could easily find that the structural modification site of **BA**, and linked with different nitrogen heterocyclic rings on **BA**, had an effect on the antitumor activity of the **BA** derivatives in vitro. In general, as observation for compounds **7a**, **7c**, **7d**, **7e** and **5a**–**5d**, structural modification at positions C-28 and C-3 could improve antitumor biological activity, and especially the structural transformation of C-28 might have more potential to enhance cytotoxicity on the same series of tumor cells; Besides, different nitrogen heterocyclic rings on **BA** also influenced their activity (compound **11e** > compounds **11a**, **11b**, **11c**, **11d**), and the alkalinities of the different nitrogen heterocyclic rings were positively correlated with their activities, which might be likely associated with increasing bioavailability and altering an extracellular weak acidic microenvironment with further verification [25].

## 4. Materials and Methods

### 4.1. Materials and Instruments

Betulinic acid (Nanjing Jingzhu Bio-technology Co., Ltd., Nanjing, China), 1-Ethyl-3-(3-dimethylaminopropyl)carbodiimide (EDCI), (Bellen Chemistry Co., Ltd., Beijing, China), 4-dimethylaminopyridine (DMAP, which is (CH_3_)_2_NC_5_H_4_N), Cyclopentylamine, Cyclohexylamine, Pyrrolidine, Piperidine, Piperazine, Succinic anhydride, Benzyl bromide, Palladium, Chromium oxide, 1,2-dibromoethane (Aladdin Bio-Chem Technology Co., Ltd., Shanghai, China), HOBt (Beijing Inno Chem Science and Technology Co., Ltd., Beijing, China) were more than 98%. All reagents were used without any further purification. Reagents of analytical reagent grade were purchased from the Beijing Chemical Plant (Beijing, China). Reactions were monitored by thin-layer chromatography (TLC) on precoated silica gel GF-254 plates (Qingdao Haiyang Chemical Co., Qingdao, China) and visualized in ultraviolet (UV) light (254 nm). Silica-gel column chromatography was performed using 200–300 mesh silica gel.

Hydrogen protonic nuclear magnetic resonance (^1^H-NMR) and Carbon-13 nuclear magnetic resonance (^13^C-NMR) assays were recorded on a Bruker AVANCE 500 NMR spectrometer (Fällanden, Switzerland). High-resolution mass spectra (HR-MS) were acquired using a Thermo Scientific TMLTQ Orbitrap XL hybrid FTMS instrument (Thermo Technologies, New York, NY, USA). Melting points were measured at a rate of 5 °C/min using an X-5 micro melting point apparatus (Beijing Tech Instrument Co., Ltd., Beijing, China). Cellular morphologies were observed using an inverted fluorescence microscope (Olympus IX71, Tokyo, Japan). Mechanisms of apoptosis were detected by flow cytometry (BD FACS Canto II, San Jose, CA, USA).

### 4.2. Chemical Syntheses

Benzyl lup-20(29)-en-28-oate (**2**). **BA** (3.00 g, 6.57 mmol) was dissolved in dimethylformamide (DMF) (200 mL), then benzyl bromide (1.12 g, 6.50 mmol), K_2_CO_3_ (2.72 g, 9.20 mmol) were added, and the mixture was stirred for 4 h at 85 °C. This reaction was monitored by TLC. The reaction solution was washed with water, filtered and evaporated with vacuum.

*Benzyl 3β-(succinic anhydride)-lup-20(29)-en-28-oate* (**3**). *Benzyl lup-20(29)-en-28-oate* (**2**) (3.00 g, 5.61 mmol), succinic anhydride (1.68 g, 16.83 mmol) and DMAP (1.37 g, 11.22 mmol) were dissolved in DCM, and then the mixture was refluxed and stirred for 8 h at 50 °C. After completion of the reaction, the crude product was extracted with DCM. After drying the organic layer over anhydrous Na_2_SO_4_ and evaporating the solvent under vacuum, the crude product was separated by flash chromatography with petroleum ether–acetone (10:1) as the eluent, then the product was lyophilized. White solid, 79.3% yield, ^1^H-NMR (500 MHz, CDCl_3_): *δ* 7.36-7.26 (m, 5H, –C_6_H_5_), 5.16–5.08 (m, 2H, –O–CH_2_–Ph), 4.72, 4.59 (brs, each, 1H, =CH_2_), 4.50–4.47 (m, 1H, –CH–O–), 2.68–2.66, 2.63–2.62 (m, each, 2H, –COO–CH_2_–CH_2_–COO–), 2.50–1.00 (28 H, methyl- and methylene- of **BA**), 1.67, 0.93, 0.82, 0.82, 0.75 (s, each, 3H, 5 × –CH_3_, methyl of **BA**); ^13^C-NMR (125 MHz, CDCl_3_): *δ* 177.98 (–COOH), 175.97 (–COO–), 171.97 (–COO–), 150.68 (–CH=C), 109.77 (–CH=C), 81.71 (–OCOCH–), 65.86, 56.67, 55.56, 50.57, 49.56, 47.08, 42.52, 40.79, 38.49, 38.31, 37.97, 37.21, 37.07, 34.34, 32.23, 30.69, 29.69, 29.45, 29.14, 28.01, 25.61, 23.75, 21.02, 19.47, 18.29, 16.63, 16.30, 15.96, 14.78; benzene ring: 136.62, 128.62, 128.38, 128.19. m.p.: 153.6–155.4 °C. HR-MS (ESI) *m*/*z*: 647.4317 [M + H]^+^, calcd for: C_41_H_59_O_6_: 647.4233.

Compound **4a**–**4d**. *Benzyl 3β-(succinic anhydride)-lup-20(29)-en-28-oate* (**3**) (0.30 g, 0.48 mmol), cyclohexylamine (63.36 g, 0.64 mmol)/cyclopentylamine (54.50 g, 0.64 mmol)/piperidine (54.50 g, 0.64 mmol)/pyrrolidine (48.07 g, 0.64 mmol), EDCI (122.69 g, 0.64 mmol), HoBt (86.86 g, 0.64 mmol) and DIPEA (82.72 g, 0.64 mmol) were dissolved in 10 mL dry DCM, the reaction mixture was stirred for 4 h at room temperature. After completion of the reaction, the crude product was extracted with DCM. After drying the organic layer over anhydrous Na_2_SO_4_ and evaporating the solvent under vacuum, the crude product was separated by flash chromatography with petroleum ether–acetone (8:1) as eluent, the product was lyophilized.

*Benzyl 3β-4-cyclohexylamino-succinic anhydride)-lup-20(29)-en-28-oate* (**4a**). White solid, 85.3% yield, 1H-NMR (500 MHz, CDCl_3_): *δ* 7.36–7.26 (m, 5H, –C6H5), 5.15- 5.10 (m, 2H, –O–CH2–Ph), 4.71, 4.59 (brs, each, 1H, =CH2), 4.49–4.46 (m, 1H, –CH–O–CO–), 3.03–2.98 (m, 1H, –N–CH–(CH2)2–), 2.67–2.63, 2.45–2.42 (m, each, 2H, –COO–CH2–CH2–COO–), 2.50–1.00 (38 H, methyl- and methylene- of **BA** and cyclohexane), 1.67, 0.93, 0.81, 0.81, 0.75 (s, each, 3H, 5 × –CH3, methyl of **BA**); 13C-NMR (125 MHz, CDCl3): δ 175.83 (–COO–), 172.91 (–COO–), 170.52 (–CO–NH–), 150.58 (–CH=C), 109.65 (–CH=C), 81.38 (–OCOCH–), 65.74 (–O–CH2–Ph), 56.56, 55.45, 50.48, 49.46, 48.21, 46.97, 42.40, 40.68, 38.40, 38.19, 37.88, 37.10, 36.96, 34.24, 33.14 (–N–CH2–C), 32.12, 31.52, 30.59, 30.23, 29.57, 28.00, 26.94, 25.56, 25.51, 24.83, 23.72, 20.91, 19.37, 18.18, 16.56, 16.20, 15.84, 14.66; benzene ring: 136.51, 128.51, 128.27, 128.08. m.p.: 145.5–147.8 °C. HR-MS (ESI) *m*/*z*: 728.5244 [M + H]^+^, calcd for: C41H59O6: 728.5176. 

*Benzyl 3β-(4-cyclohexylamine-succinic anhydride)-lup-20(29)-en-28-oate* (**4b**). White solid, 82.8% yield, ^1^H-NMR (500 MHz, CDCl_3_): *δ* 7.36–7.26 (m, 5H, –C_6_H_5_), 5.16–5.07 (m, 2H, –O–CH_2_–Ph), 4.72, 4.59 (brs, each, 1H, =CH_2_), 4.49–4.46 (m, 1H, –CH–O–CO–), 3.04–2.99 (m, 1H, –N–CH–(CH_2_)_2_–), 2.65–2.64, 2.44–2.42 (m, each, 2H, –COO–CH_2_–CH_2_–COO–), 2.50–1.00 (36 H, methyl- and methylene- of **BA** and cyclopentylamine), 1.67, 0.93, 0.81, 0.81, 0.75 (s, each, 3H, 5 × –CH_3_, methyl of **BA**); ^13^C-NMR (125 MHz, CDCl_3_): *δ* 175.95 (–COO–), 173.04 (–COO–), 171.14 (–CO–NH–), 150.70 (–CH=C), 109.76 (–CH=C), 81.52 (–OCOCH–), 65.86, 56.67, 55.56, 51.35, 50.60, 49.57, 47.08, 42.52, 40.79, 38.51, 38.31, 37.99, 37.22, 37.07, 34.35, 33.25, 33.23, 32.24, 31.55, 30.70, 30.34, 29.69, 28.11, 25.63, 23.83, 21.02, 19.48, 19.33, 18.30, 16.67, 16.31, 15.96, 14.77, 13.88; benzene ring: 136.63, 128.62, 128.38, 128.19. m.p.: 147.2-149.6 °C. HR-MS (ESI) *m*/*z*: 714.5098 [M + H]^+^, calcd for: C_41_H_59_O_6_ 714.5019.

*Benzyl 3β-(4-pyrrolidine-succinic anhydride)-lup-20(29)-en-28-oate* (**4c**). White solid, 78.8% yield, ^1^H-NMR (500 MHz, CDCl_3_): *δ* 7.36-7.26 (m, 5H, –C_6_H_5_), 5.13, −5.10 (m, 2H, –O–CH_2_–Ph), 4.71 (brs, each, 1H, =CH_2_), 4.59, 4.48–4.45 (m, 1H, –CH–O–CO–), 2.68–2.56 (m, 4H, –COO–CH_2_–CH_2_–COO–), 2.50–1.00 (36 H, methyl- and methylene- of **BA** and piperdine), 1.67, 0.93, 0.83, 0.81, 0.75 (s, each, 3H, 5 × -CH_3_, methyl of **BA**); ^13^C-NMR (125 MHz, CDCl_3_): *δ* 175.95 (–COO–), 173.10 (–COO–), 169.93 (–CO–NH–), 150.69 (–CH=C), 109.76 (–CH=C), 81.19 (–OCOCH–), 65.85, 56.67, 55.58, 50.58, 49.57, 47.09, 42.52, 40.79, 38.51, 38.32, 37.98, 37.21, 37.07, 34.36, 32.24, 30.70, 29.69, 29.64, 29.56, 28.10, 26.20, 25.63, 23.80, 21.02, 19.47, 18.29, 16.63, 16.30, 15.95, 14.79; benzene ring: 136.63, 128.62, 128.37, 128.18. m.p.: 135.0–138.6 °C. HR-MS (ESI) *m*/*z*: 714.5079 [M + H]^+^, calcd for: C_41_H_59_O_6_ 714.5019. 

*Benzyl 3β-(4- piperidine -succinic anhydride)-lup-20(29)-en-28-oate* (**4d**). White solid, 80.8% yield, ^1^H-NMR (500 MHz, CDCl_3_): *δ* 7.36–7.26 (m, 5H, –C_6_H_5_), 5.13–5.10 (m, 2H, –O–CH_2_–Ph), 4.72 (brs, each, 1H, =CH_2_), 4.59, 4.48–4.45 (m, 1H, –CH–O–CO–), 2.64–2.62, 2.62–2.59 (–COO–CH_2_–CH_2_–COO), 2.50–1.00 (34 H, methyl- and methylene- of **BA** and pyrrolidine), 1.67, 0.93, 0.83, 0.82, 0.75 (s, each, 3H, 5 × –CH_3_, methyl of **BA**); ^13^C-NMR (125 MHz, CDCl_3_): *δ* 175.94 (-COO-), 173.09 (-COO-), 169.53 (–CO–NH–), 150.68 (–CH=C), 109.75 (–CH=C), 81.09 (–OCOCH–), 65.84, 56.66, 55.58, 50.57, 49.56, 47.08, 42.51, 40.79, 38.50, 38.31, 37.98, 37.21, 37.06, 34.35, 32.23, 30.69, 29.98, 29.68, 28.19, 28.08, 26.45, 25.62, 23.79, 21.01, 19.46, 18.28, 16.68, 16.29, 15.95, 14.78; benzene ring: 136.62, 128.61, 128.37, 128.17. m.p.: 132.1–135.4 °C. HR-MS (ESI) *m*/*z*: 700.4936 [M + H]^+^, calcd for: C_41_H_59_O_6_ 700.4863.

Compound **5a**–**5d**. Compound **4a**/**4b**/**4c**/**4d** (0.20 g, 0.46 mmol) was dissolved in dry MEOH, then suitable palladium carbon was added. The reaction mixture was stirred overnight in methanol in a hydrogen atmosphere. The reaction was monitored by TLC [petroleum ether–acetone (4:1)]. After filtering the palladium carbon and evaporating the solvent under vacuum, the crude product was separated by flash chromatography with petroleum ether-acetone (8:1) as eluent, the product was lyophilized.

*3β-(4-cyclohexylamino- succinic anhydride)-lup-20(29)-en-28-oate* (**5a**). White solid, 85.8% yield, ^1^H-NMR (500 MHz, CDCl_3_): *δ* 4.72, 4.59 (brs, each, 1H, =CH_2_), 4.46–4.50 (m, 1H, –CH–O–CO–), 3.02–2.97 (m, 1H, –N–CH–(CH_2_)_2_–), 2.65–2.63, 2.45–2.42 (m, each, 2H, –COO–CH_2_–CH_2_–COO–), 2.50–1.00 (38 H, methyl- and methylene- of **BA** and cyclohexane), 1.68, 0.96, 0.92, 0.83, 0.81 (s, each, 3H, 5 × –CH_3_, methyl of **BA**); ^13^C-NMR (125 MHz, CDCl_3_): *δ* 180.79 (–COOH), 173.13 (–COO–), 170.87 (–CO–NH–), 150.61 (–CH=C), 109.80 (–CH=C), 81.58 (–OCOCH–), 56.44, 55.55, 50.92, 50.54, 49.37, 48.38, 47.06, 42.55, 40.82, 38.48, 37.99, 37.23, 37.19, 34.36, 33.18, 32.31, 31.60, 30.70, 30.35, 29.81, 28.10, 25.64, 25.58, 24.91, 23.81, 21.00, 19.47, 18.28, 16.64, 16.29, 16.15, 14.77. m.p.: 191.3–193.0 °C. HR-MS (ESI) *m*/*z*: 638.4773 [M + H]^+^, calcd for: C_41_H_59_O_6_ 638.4706.

*3β-(4-cyclohexylamine- succinic anhydride)-lup-20(29)-en-28-oate* (**5b**). White solid, 87.2% yield, ^1^H-NMR (500 MHz, CDCl_3_): *δ* 4.73, 4.60 (brs, each, 1H, =CH_2_), 4.49–4.46 (m, 1H, –CH–O–CO–), 3.02–2.97 (m, 1H, –N–CH–(CH_2_)_2_–), 2.69–2.67, 2.58–2.56 (m, each, 2H, –COO–CH_2_–CH_2_–COO–), 2.50–1.00 (36 H, methyl- and methylene- of **BA** and cyclopentylamine), 1.68, 0.96, 0.92, 0.84, 0.82 (s, each, 3H, 5 × –CH_3_, methyl of **BA**); ^13^C-NMR (125 MHz, CDCl_3_): *δ* 181.68 (–COOH), 173.10 (–COO–), 170.17 (–CO–NH–), 150.57 (–CH=C), 109.84 (–CH=C), 81.25 (–OCOCH–), 56.50, 55.58, 50.53, 49.39, 47.06, 46.72, 42.56, 40.83, 38.52, 37.99, 37.54, 37.25, 34.37, 32.30, 31.07, 30.70, 29.83, 29.65, 29.58, 28.11, 26.18, 25.59, 24.55, 23.80, 20.99, 19.48, 18.28, 16.62, 16.29, 16.16, 14.81. m.p.: 247.3–249.6 °C. HR-MS (ESI) *m*/*z*: 624.4664 [M + H]^+^, calcd for: C_41_H_59_O_6_ 624.4550. 

*3β-(4-pyrrolidine-succinic anhydride)-lup-20(29)-en-28-oate* (**5c**). White solid, 89.8% yield, ^1^H-NMR (500 MHz, CDCl_3_): *δ* 4.73, 4.60 (brs, each, 1H, =CH_2_), 4.50–4.47 (m,1H,–CH–O–CO–), 3.03–2.97 (m, 4H, –N–(CH_2_)_2_–(CH_2_)_2_), 2.67–2.64 (m, 4H, –COO–CH_2_–CH_2_–COO–), 2.50–1.00 (34 H, methyl- and methylene- of **BA** and pyrrolidine), 1.69, 0.96, 0.93, 0.84, 0.82 (s, each, 3H, 5 × –CH_3_, methyl of **BA**); ^13^C-NMR (125 MHz, CDCl_3_): *δ* 181.67 (–COOH), 173.10 (–COO–), 171.31 (–CO–NH–), 150.56 (–CH=C), 109.85 (–CH=C), 81.56 (–OCOCH–), 56.49, 55.56, 51.38, 50.55, 49.39, 47.07, 42.56, 40.84, 38.52, 38.01, 37.25, 37.20, 34.37, 33.99, 33.23, 33.22, 32.30, 31.55, 30.71, 30.35, 29.83, 28.12, 25.59, 23.84, 21.00, 19.49, 18.29, 16.66, 16.31, 16.17, 14.80. m.p.: 231.4–233.8 °C. HR-MS (ESI) *m*/*z*: 624.4611 [M + H]^+^, calcd for: C_41_H_59_O_6_ 624.4550.

*3β-(4- piperidine -succinic anhydride)-lup-20(29)-en-28-oate* (**5d**). White solid, 88.6% yield, ^1^H-NMR (500 MHz, CDCl_3_): *δ* 4.73, 4.60 (brs, each, 1H, =CH_2_), 4.5–4.47 (m,1H,–CH–O–CO–), 3.03–2.97 (m,4H,–N–(CH_2_)_2_–(CH_2_)_2_), 2.67–2.64, 2.45–2.43 (m, each, 2H,–COO–CH_2_–CH_2_–COO–), 2.50–1.00 (32 H, methyl- and methylene- of **BA** and piperdine), 1.69, 0.96, 0.93, 0.84, 0.82 (s, each, 3H, 5 × –CH_3_, methyl of **BA**); ^13^C-NMR (125 MHz, CDCl_3_): *δ* 181.67 (–COOH), 173.10 (–COO–), 171.31 (–CO–NH–), 150.56 (–CH=C), 109.85 (–CH=C), 81.56 (–OCOCH–), 56.49, 55.56, 51.38, 50.55, 49.39, 47.07, 42.56, 40.84, 38.52, 38.01, 37.25, 37.20, 34.37, 33.23, 33.22, 32.30, 31.55, 30.71, 30.35, 29.83, 28.12, 25.59, 23.84, 21.00, 19.49, 18.29, 16.66, 16.31, 16.17, 14.80. m.p.: 238.2–240.7 °C. HR-MS (ESI) *m*/*z*: 624.4611 [M + H]^+^, calcd for: C_41_H_59_O_6_ 624.4550.

*1-bromopropane lup-20(29)-en-28-oate* (**6**). **BA** (8.00 g, 17.52 mmol) was dissolved in DMF (300 mL), and then 1, 2-dibromoethane (9.80 g, 52.56 mmol) and K_2_CO_3_ (4.84 g, 35.04 mmol) were added, and the mixture was stirred for 2 h at room temperature. Reaction was monitored by TLC [petroleum ether–acetone (5:1)]. After completion of the reaction, the crude product was extracted with EtOAc. After drying, the organic layer over anhydrous Na_2_SO_4_ and evaporating the solvent under vacuum, the crude product was separated by flash chromatography with petroleum ether–acetone (50:1) as eluent, the product was lyophilized. White solid, 48.7% yield, ^1^H-NMR (400 MHz, CDCl_3_): *δ* 4.73, 4.60 (brs, each, 1H, =CH_2_), 4.42–4.38, 3.55–3.52 (m, each, 2H, –CO–CH_2_–CH_2_–Br), 3.20–3.16 (m, 1H, –(CH_2_)_2_–CH–OH), 2.50–1.00 (28 H, methyl- and methylene- of **BA**), 1.68, 0.96, 0.91, 0.81, 0.75 (s, each, 3H, 5 × –CH_3_, methyl of **BA**); ^13^C-NMR (100 MHz, CDCl_3_): *δ* 175.87 (–COO–), 150.58 (–CH=C), 109.82 (–CH=C), 79.11 (CH–OH), 63.48, 56.83, 55.49, 50.69, 49.56, 47.09, 42.55, 40.88, 39.00, 38.87, 38.48, 37.33, 37.13, 34.45, 32.20, 30.73, 29.82, 29.30, 28.12, 27.55, 25.67, 21.03, 19.51, 18.43, 16.27, 16.14, 15.50, 14.86. m.p.: 194.2–196.7 °C. HR-MS (ESI) *m*/*z*: 563.3105 [M + H]^+^, calcd for: C_32_H_52_BrO_3_ 563.3022. 

*1-bromopropane 3-oxolup-20(29)-en-28-oate* (**8**). 1-bromopropane lup-20(29)-en-28-oate (**6**) (2.00 g, 3.56 mmol) was dissolved in acetone (150 mL), then chromic acid was added uniformly to the acetone solution, and the mixture was stirred for 1 h at 0 °C. Reaction was monitored by TLC [petroleum ether–acetone (5:1)]. After completion of the reaction, the crude product was extracted with EtOAc. After drying the organic layer over anhydrous Na_2_SO_4_ and evaporating the solvent under vacuum, the product was lyophilized. White solid, 90.6% yield, ^1^H-NMR (400 MHz, CDCl_3_): *δ* 4.74, 4.61 (brs, each, 1H, =CH_2_). 4.38–4.43, 3.52–3.55 (m, each, 2H, –CO–CH_2_–CH_2_–Br), 1.00–2.50 (28 H, methyl- and methylene- of **BA**), 1.68, 1.06, 0.98, 0.96, 0.92 (s, each, 3H, 5 × –CH_3_, methyl of **BA**); ^13^C-NMR (100 MHz, CDCl_3_): *δ* 218.28 (C=O), 175.84 (–COO–), 150.50 (–CH=C), 109.88 (–CH=C), 63.51, 56.81, 55.12, 50.05, 49.49, 47.48, 47.06, 42.61, 40.83, 39.78, 38.56, 37.11, 37.05, 34.29, 33.75, 32.14, 30.71, 29.80, 29.32, 26.76, 25.68, 21.56, 21.17, 19.78, 19.51, 16.10, 15.96, 14.77. m.p.: 140.9–142.6 °C. HR-MS (ESI) *m*/*z*: 561.2958 [M + H]^+^, calcd for: C_32_H_50_BrO_3_ 561.2865. 

*Benzyl 3β-(2-chloroacetic acid**)-lup-20(29)-en-28-oate* (**10**). *Benzyl lup-20(29)-en-28-oate* (**2**) (3.00 g, 5.61 mmol), chloroacetic acid (1.06 g, 11.22 mmol) and DMAP (1.37 g, 11.22 mmol) was dissolved in DCM, then EDCI (2.15 g, 11.22 mmol) was added after 5 min. Reaction was monitored by TLC [petroleum ether–acetone (5:1)]. After completion of the reaction, the crude product was extracted with 10% HCl three times, washing with water three times subsequently. After drying the organic layer over anhydrous Na_2_SO_4_ and evaporating the solvent under vacuum, the crude product was separated by flash chromatography with petroleum ether-acetone (10:1) as eluent, the product was lyophilized. White solid, 80.2% yield, ^1^H-NMR (400 MHz, CDCl_3_): *δ* 7.30–7.37 (m, 5H, –C_6_H_5_), 5.07–5.16 (m, 2H, –O–CH_2_–Ph), 4.72, 4.60 (brs, each, 1H, =CH_2_), 4.53–4.57 (m, 1H, –CH–O–), 4.00–4.08 (m, 2H, Cl–CH_2_–CO–), 1.00–2.50 (28 H, methyl- and methylene- of **BA**), 1.68, 0.94, 0.86, 0.85, 0.76 (s, each, 3H, 5 × –CH_3_, methyl of **BA**); ^13^C-NMR (100 MHz, CDCl_3_): *δ* 175.94 (–COO–), 167.27 (–COO–), 150.67 (–CH=C), 109.78 (–CH=C), 83.53 (C–OH), 65.87, 56.69, 55.53, 50.60, 49.59, 47.10, 42.55, 41.39, 40.82, 38.47, 38.32, 38.16, 37.23, 37.07, 34.35, 32.25, 30.72, 29.70, 28.07, 25.62, 23.70, 21.05, 19.49, 18.28, 16.56, 16.32, 15.98, 14.78. benzene ring: 136.64, 128.63, 128.39, 128.19. m.p.: 144.7–146.4 °C. HR-MS (ESI) *m*/*z*: 623.3882 [M + H]^+^, calcd for: C_33_H_60_ClO_4_ 623.3789. 

Compound **7a**–**7e**. 1-bromopropane lup-20(29)-en-28-oate (**6**) (0.30 g, 0.53 mmol) and cyclohexylamine (262.35 g, 2.65 mmol)/piperidine (135.15 g, 1.59 mmol)/pyrrolidine (11.08 g, 1.59 mmol)/piperazine (54.50 g, 0.64 mmol) were dissolved in 20 mL DMF, then K_2_CO_3_ (146.50 g, 1.06 mmol) was added after 5 min. The reaction mixture was stirred at room temperature overnight. After completion of the reaction, the crude product was extracted with EtOAc. After drying the organic layer over anhydrous Na_2_SO_4_ and evaporating the solvent under vacuum, the crude product was separated by flash chromatography with petroleum ether–acetone (50:3) as eluent, the product was lyophilized.

*N-propylcyclohexanamine lup-20(29)-en-28-oate* (**7a**). White solid, 35.8% yield, ^1^H-NMR (400 MHz, CDCl_3_): *δ* 4.72, 4.59 (brs, each, 1H, =CH_2_), 4.24–4.14, 2.90–2.87 (m, each, 2H, –CO–CH_2_–CH_2_–Br), 3.20–3.16 (m, 1H, –(CH_2_)_2_–CH–OH), 2.50–1.00 (38 H, methyl- and methylene- of **BA** and cyclohexane), 1.68, 0.96, 0.91, 0.81, 0.75 (s, each, 3H, 5 × –CH_3_, methyl of **BA**); ^13^C-NMR (100 MHz, CDCl_3_): *δ* 176.17 (–COO–), 150.63 (–CH=C), 109.78 (–CH=C), 79.10 (CH–OH), 63.79, 56.76, 56.52, 55.49, 50.67, 49.54, 47.24, 45.53, 42.57, 40.83, 39.00, 38.85, 38.49, 37.32, 34.47, 33.74, 33.69, 32.41, 30.79, 29.82, 28.12, 27.54, 26.23, 25.67, 25.12, 21.02, 19.51, 18.42, 16.24, 16.21, 15.50, 14.84. m.p.: 119.0–121.8 °C. HR-MS (ESI) *m*/*z*: 582.4917 [M + H]^+^, calcd for: C_38_H_64_NO_3_ 582.4808.

*1-propylpiperidine lup-20(29)-en-28-oate* (**7c**). White solid, 53.3% yield, ^1^H-NMR (400 MHz, CDCl_3_): *δ* 4.72, 4.59 (brs, each, 1H, =CH_2_), 4.28–4.20, 3.04–2.98 (m, each, 2H, –CO–CH_2_–CH_2_–N–), 3.19–3.15 (m, 1H, –(CH_2_)_2_–CH–OH), 2.50–1.00 (38 H, methyl- and methylene- of **BA** and pyrrolidine), 1.68, 0.96, 0.91, 0.81, 0.75 (s, each, 3H, 5 × –CH_3_, methyl of **BA**); ^13^C-NMR (100 MHz, CDCl_3_): *δ* 176.16 (–COO–), 150.82 (–CH=C), 109.67 (–CH=C), 79.12 (CH–OH), 61.60, 57.58, 56.67, 55.51, 54.89, 50.71, 49.55, 47.11, 42.56, 40.88, 39.01, 38.88, 38.40, 37.35, 37.18, 34.50, 32.32, 30.78, 29.80, 28.13, 27.57, 26.13, 25.70, 24.35, 21.05, 19.53, 18.46, 16.27, 16.21, 15.50, 14.84. m.p.: 138.6–140.5 °C. HR-MS (ESI) *m*/*z*: 568.4743 [M + H]^+^, calcd for: C_37_H_62_NO_3_ 568.4651.

*1-propylpyrrolidine lup-20(29)-en-28-oate* (**7d**). White solid, 58.1% yield, ^1^H-NMR (400 MHz, CDCl_3_): *δ* 4.72, 4.59 (brs, each, 1H, =CH_2_), 4.25–4.21, 3.01–2.97 (m, each, 2H, –CO–CH_2_–CH_2_–N–), 3.19–3.15 (m, –OH), 2.50–1.00 (36 H, methyl- and methylene- of **BA** and piperdine), 1.67, 0.95, 0.91, 0.81, 0.75 (s, each, 3H, 5 × –CH_3_, methyl of **BA**); ^13^C-NMR (100MHz, CDCl_3_): *δ* 176.06 (–COO–), 150.75 (–CH=C), 109.68 (–CH=C), 79.10 (CH–OH), 62.72, 56.64, 55.50, 54.56, 54.45, 50.71, 49.56, 47.05, 42.54, 40.86, 39.00, 38.87, 38.35, 37.33, 37.12, 34.48, 32.26, 30.73, 29.79, 28.13, 27.55, 25.69, 23.68, 21.03, 19.52, 18.44, 16.27, 16.16, 15.50, 14.83. m.p.: 160.7–162.4 °C. HR-MS (ESI) *m*/*z*: 554.4544 [M + H]^+^, calcd for: C_36_H_59_NO_3_ 554.4495. 

*1-propylpiperazine lup-20(29)-en-28-oate* (**7e**). White solid, 30.2% yield, ^1^H-NMR (400 MHz, CDCl_3_): *δ* 4.71, 4.59 (brs, each, 1H, =CH_2_), 4.19–4.22, 2.91–3.01, (m, each, 2H, –CO–CH_2_–CH_2_–N–), 3.16–3.19 (m, 1H, –(CH_2_)_2_–CH–OH), 2.62–2.64 (m, 4H, NH–(CH_2_)_2_–), 1.00–2.50 (32 H, methyl- and methylene- of **BA** and piperazine). 1.67, 0.95, 0.90, 0.81, 0.74 (s, each, 3H, 5 × –CH_3_, methyl of **BA**); ^13^C-NMR (100 MHz, CDCl_3_): *δ* 176.07 (–COO–), 150.62 (–CH=C), 109.77 (–CH=C), 79.08 (CH–OH), 60.87, 57.09, 56.67, 55.49, 52.50, 50.67, 49.50, 47.10, 44.98, 42.54, 40.85, 38.99, 38.85, 38.37, 37.33, 37.15, 34.49, 32.25, 30.73, 29.77, 28.13, 27.54, 25.65, 21.03, 19.49, 18.43, 16.27, 16.22, 15.51, 14.82. m.p.: 197.9–199.2 °C. HR-MS (ESI) *m*/*z*: 569.4687 [M + H]^+^, calcd for: C_36_H_61_N_2_O_3_ 569.4604. 

Compound **9a**–**9d**. 1-bromopropane 3-oxolup-20(29)-en-28-oate (**8**) (0.30 g, 0.53 mmol) and cyclohexylamine (262.35 g, 2.65 mmol)/cyclopentylamine (225.25 g, 2.65 mmol)/piperidine (135.15 g, 1.59 mmol)/pyrrolidine (113.08 g, 1.59 mmol)/piperazine (341.11 g, 3.18 mmol) were dissolved in 20 mL DMF, then K_2_CO_3_ (146.50 g, 1.06 mmol) was added after 5 min. The reaction mixture was stirred at room temperature overnight. Reaction was monitored by TLC [petroleum ether–acetone (5:1)]. After completion of the reaction, the crude product was extracted with EtOAc. After drying the organic layer over anhydrous Na_2_SO_4_ and evaporating the solvent under vacuum, the crude product was separated by flash chromatography with petroleum ether–acetone (25:1) as eluent, the product was lyophilized. 

*N-propylcyclohexanamine 3-oxolup-20(29)-en-28-oate* (**9a**). White solid, 38.4% yield, ^1^H-NMR (400 MHz, CDCl_3_): *δ* 4.73, 4.60 (brs, each, 1H, =CH_2_), 2.88–2.92, 4.17–4.29 (m, each, 2H, –CO–CH_2_–CH_2_–N–), 2.47–2.52 (m, 1H, –N–CH–(CH_2_)_2_), 1.00–2.50 (38 H, methyl- and methylene- of **BA** and cyclohexane), 1.68, 1.06, 1.02, 0.97, 0.92 (s, each, 3H, 5 × –CH_3_, methyl of **BA**); ^13^C-NMR (125 MHz, CDCl_3_): *δ* 218.23 (C=O), 176.15 (–COO-), 150.55 (–CH=C), 109.86 (–CH=C), 63.67, 56.74, 56.59, 55.16, 50.05, 49.50, 47.49, 47.19, 45.44, 42.65, 40.81, 39.78, 38.56, 37.20, 37.06, 34.30, 33.80, 33.50, 32.33, 30.77, 29.81, 26.74, 26.18, 25.69, 25.09, 21.56, 21.19, 19.78, 19.52, 16.07, 16.03, 14.77. m.p.: 142.8–144.1 °C. HR-MS (ESI) *m*/*z*: 580.4701 [M + H]^+^, calcd for: C_38_H_62_NO_3_ 580.4651. 

*N-propylcyclopentanamine 3-oxolup-20(29)-en-28-oate* (**9b**). White solid, 35.2% yield, ^1^H-NMR (400 MHz, CDCl_3_): *δ* 4.73, 4.61 (brs, each, 1H, =CH_2_), 4.29–4.32, 2.93–3.05 (m, each, 2H, –CO–CH_2_–CH_2_–N–), 2.40–2.49 (m, 1H, –N–CH–(CH_2_)_2_), 1.00–2.50 (36 H, methyl- and methylene- of **BA** and cyclopentylamine), 1.68, 1.07, 1.02, 0.97, 0.92 (s, each, 3H, 5 × –CH_3_, methyl of **BA**); ^13^C-NMR (125 MHz, CDCl_3_): *δ* 218.23 (C=O), 176.08 (–COO–), 150.40 (–CH=C), 109.95 (–CH=C), 65.72, 59.63, 56.73, 55.14, 50.05, 49.50, 47.49, 47.05, 46.45, 42.62, 40.80, 39.78, 38.49, 37.06, 34.29, 33.78, 32.10, 30.72, 30.68, 29.85, 26.75, 25.66, 24.09, 21.56, 21.19, 19.78, 19.50, 19.34, 16.10, 15.98, 14.76, 14.27, 13.88. m.p.: 145.0–147.9 °C. HR-MS (ESI) *m*/*z*: 566.4566 [M + H]^+^, calcd for: C_37_H_60_NO_3_ 566.4495

*1-propylpiperidine 3-oxolup-20(29)-en-28-oate* (**9c**). White solid, 56.6% yield, ^1^H-NMR (400 MHz, CDCl_3_): *δ* 4.72, 4.59 (brs, each, 1H, =CH_2_), 4.20–4.29, 2.98–3.07 (m, each, 2H, –CO–CH_2_–CH_2_–N–), 1.00–2.50 (38 H, methyl- and methylene- of **BA** and pyrrolidine), 1.68, 1.06, 1.01, 0.96, 0.92 (s, each, 3H, 5 × –CH_3_, methyl of **BA**); ^13^C-NMR (125 MHz, CDCl_3_): *δ* 218.27 (C=O), 176.10 (–COO–), 150.72 (–CH=C), 109.73 (–CH=C), 61.58, 57.56, 56.63, 55.12, 54.88, 50.05, 49.46, 47.47, 47.06, 42.60, 40.81, 39.77, 38.45, 37.13, 37.05, 34.29, 33.78, 32.23, 30.73, 29.76, 26.75, 26.09, 25.69, 24.31, 21.56, 21.17, 19.79, 19.52, 16.09, 16.00, 14.75. m.p.: 135.0–137.4 °C. HR-MS (ESI) *m*/*z*: 566.4595 [M + H]^+^, calcd for: C_37_H_60_NO_3_ 566.44949.

*1-propylpyrrolidine 3-oxolup-20(29)-en-28-oate* (**9d**). White solid, 59.0% yield, ^1^H-NMR (400 MHz, CDCl_3_): *δ* 4.72, 4.60 (brs, each, 1H, =CH_2_), 4.20–4.32, 2.99–3.06 (m, each, 2H, –CO–CH_2_–CH_2_–N–),1.00–2.50 (38 H, methyl- and methylene- of **BA** and pyrrolidine), 1.68, 1.06, 1.01, 0.97, 0.92 (s, each, 3H, 5 × –CH_3_, methyl of **BA**); ^13^C-NMR (125 MHz, CDCl_3_): *δ* 218.28 (C=O), 176.05 (–COO–), 150.70 (–CH=C), 109.74 (–CH=C), 56.62, 55.13, 54.67, 54.56, 50.07, 49.49, 47.48, 47.03, 42.61, 40.80, 39.78, 38.43, 37.11, 37.05, 34.29, 33.78, 32.21, 30.72, 29.76, 26.76, 25.70, 23.72, 21.57, 21.17, 19.79, 19.52, 16.10, 15.97, 14.75. m.p.: 120.1–122.6 °C. HR-MS (ESI) *m*/*z*: 552.4468 [M + H]^+^, calcd for: C_36_H_58_NO_3_ 552.4338. 

Compound **11a**–**11e**. Benzyl 3β-(2-chloroacetic acid)-lup-20(29)-en-28-oate (**10**) (0.30 g, 0.48 mmol) and cyclohexylamine (237.60 g, 2.40 mmol)/cyclopentylamine (204.00 g, 2.40 mmol)/piperidine (122.40 g, 1.44 mmol)/pyrrolidine (102.41 g, 1.44 mmol)/piperazine (248.08 g, 2.88 mmol) were dissolved in 20 mL DMF, then K_2_CO_3_ (132.68 g, 0.96 mmol) was added after 5 min. The reaction mixture was stirred at room temperature overnight. Reaction was monitored by TLC [petroleum ether–acetone (5:1)]. After completion of the reaction, the crude product was extracted with EtOAc. After drying the organic layer over anhydrous Na_2_SO_4_ and evaporating the solvent under vacuum, the crude product was separated by flash chromatography with petroleum ether–acetone (100:3) as eluent, the product was lyophilized.

*Benzyl 3β-cyclohexylglycine-lup-20(29)-en-28-oate* (**11a**). White solid, 35.1% yield, ^1^H-NMR(400 MHz, CDCl_3_): *δ* 7.26–7.36 (m, 5H, -C_6_H_5_), 5.07–5.19 (m, 2H, -O-CH_2_-Ph), 4.72, 4.59 (brs, each, 1H, =CH_2_), 4.50–4.54 (m, 1H, –CH–O–), 3.42–3.46 (m, 2H, –NH–CH_2_–CO–), 1.00–2.50 (38 H, methyl- and methylene- of **BA** and cyclohexane), 1.67, 0.93, 0.82, 0.75 (s, each, 3H, 5 × –CH_3_, methyl of **BA**); ^13^C-NMR (100 MHz, CDCl_3_): *δ* 175.93 (–COO–), 172.75 (–COO–), 150.65 (–CH=C), 109.76 (–CH=C), 81.66 (C–OH), 65.84, 56.66, 56.56, 55.53, 50.57, 49.55, 48.51, 47.07, 42.51, 40.78, 38.48, 38.29, 37.97, 37.20, 37.05, 34.33, 33.37, 32.22, 31.05, 30.69, 29.82, 29.67, 28.12, 25.60, 24.96, 23.85, 21.01, 19.46, 18.28, 16.65, 16.30, 15.95, 14.76. benzene ring: 136.61, 128.60, 128.36, 128.17. m.p.: 129.1–131.4 °C. HR-MS (ESI) *m*/*z*: 686.5148 [M + H]^+^, calcd for: C_45_H_68_NO_4_ 686.50701. 

*Benzyl 3β-cyclopentylglycine-lup-20(29)-en-28-oate* (**11b**). White solid, 39.4% yield, ^1^H-NMR (400 MHz, CDCl_3_): *δ* 7.26–7.36 (m, 5H, –C_6_H_5_), 5.07–5.16 (m, 2H, –O–CH_2_–Ph), 4.72, 4.59 (brs, each, 1H, =CH_2_), 4.51–4.55 (m, 1H, –CH–O–), 3.38–3.42 (m, 2H, –NH–CH_2_–CO–), 1.00–2.50 (36 H, methyl- and methylene- of **BA** and cyclopentylamine), 1.68, 0.94, 0.83, 0.82, 0.76 (s, each, 3H, 5 × –CH_3_, methyl of **BA**); ^13^C-NMR (125 MHz, CDCl_3_): *δ* 175.94 (–COO–), 172.63 (–COO–), 150.67 (–CH=C), 109.76 (–CH=C), 81.65 (C–OH), 65.85, 59.45, 56.67, 55.54, 50.59, 50.08, 49.57, 47.09, 42.52, 40.80, 38.50, 38.31, 38.03, 37.98, 37.21, 37.06, 34.35, 33.27, 33.07, 32.23, 30.70, 29.69, 28.15, 28.04, 25.62, 24.13, 23.88, 23.60, 21.03, 19.47, 18.30, 16.66, 16.57, 16.31, 15.96, 14.77. benzene ring: 136.63, 128.61, 128.37, 128.18. m.p.: 131.3–133.8 °C. HR-MS (ESI) *m*/*z*: 672.4986 [M + H]^+^, calcd for: C_44_H_66_NO_4_ 672.4914. 

*Benzyl 3β-(2-(piperidin-1-yl)acetic acid)-lup-20(29)-en-28-oate* (**11c**). White solid, 57.2% yield, ^1^H-NMR (400 MHz, CDCl_3_): *δ* 7.26–7.36 (m, 5H, –C_6_H_5_), 5.07–5.15 (m, 2H, –O–CH_2_–Ph), 4.71, 4.59 (brs, each, 1H, =CH_2_), 4.50–4.54 (m, 1H, –CH–O–), 3.17–3.21 (m, 2H, –NH–CH_2_–CO–), 1.00–2.50 (36 H, methyl- and methylene- of **BA** and piperdine), 1.67, 0.93, 0.81, 0.75 (s, each, 3H, 5 × –CH_3_, methyl of **BA**); ^13^C-NMR (100 MHz, CDCl_3_): *δ* 175.97 (–COO–), 170.59 (–COO–), 150.67 (–CH=C), 109.76 (–CH=C), 81.30 (C–OH), 65.85, 60.45, 56.67, 55.51, 54.27, 50.95, 50.57, 49.56, 47.09, 42.52, 40.79, 38.49, 38.31, 37.93, 37.20, 37.06, 34.34, 32.23, 30.69, 29.83, 29.68, 28.15, 25.91, 25.61, 24.01, 23.95, 21.02, 19.45, 18.30, 16.71, 16.30, 15.95, 14.77. Benzene ring: 136.62, 128.61, 128.36, 128.17. m.p.: 146.8–148.0 °C. HR-MS (ESI) *m*/*z*: 672.4993 [M + H]^+^, calcd for: C_44_H_66_NO_4_ 672.49136. 

*Benzyl 3β-(2-(pyrrolidin-1-yl)acetic acid)-lup-20(29)-en-28-oate* (**11d**). White solid, 50.7% yield, ^1^H-NMR (400 MHz, CDCl_3_): *δ* 7.27–7.36 (m, 5H, –C_6_H_5_), 5.07–5.16 (m, 2H, –O–CH_2_–Ph), 4.72, 4.59 (brs, each, 1H, =CH_2_), 4.52–4.56 (m, 1H, –CH–O–), 3.29–3.38 (m, 2H, –NH–CH_2_–CO–), 1.00–2.50 (36 H, methyl- and methylene- of **BA** and pyrrolidine), 1.67, 0.93, 0.83, 0.82, 0.75 (s, each, 3H, 5 × –CH_3_, methyl of **BA**); ^13^C-NMR (100 MHz, CDCl_3_): *δ* 175.95 (–COO–), 170.79 (–COO–), 150.67 (–CH=C), 109.65 (–CH=C), 81.15 (C-OH), 65.84, 57.01, 56.67, 55.52, 54.00, 50.58, 49.57, 47.09, 42.51, 40.79, 38.51, 38.31, 37.98, 37.21, 37.07, 34.35, 32.23, 30.70, 29.69, 28.13, 25.61, 23.96, 23.91, 21.02, 19.47, 18.30, 16.70, 16.31, 15.96, 14.76. Benzene ring: 136.63, 128.61, 128.37, 128.18. m.p.: 139.8–142.6 °C. HR-MS (ESI) *m*/*z*: 658.4842 [M + H]^+^, calcd for: C_43_H_64_NO_4_ 658.4757. 

*Benzyl 3β-(2-(piperazin-1-yl)acetic acid)-lup-20(29)-en-28-oate* (**11e**). White solid, 32.6% yield, ^1^H-NMR (400 MHz, CDCl_3_): *δ* 7.29–7.36 (m, 5H, –C_6_H_5_), 5.07–5.16 (m, 2H, –O–CH_2_–Ph), 4.71, 4.59 (brs, each, 1H, =CH_2_), 4.50–4.54 (m, 1H, –CH–O–), 3.22–3.25 (m, 2H, –NH–CH_2_–CO–), 2.61–2.70 (m, 4H, NH–(CH_2_)_2_–), 1.00–2.50 (36 H, methyl- and methylene- of **BA** and pyrrolidine), 1.67, 0.93, 0.81, 0.75 (s, each, 3H, 5 × –CH_3_, methyl of **BA**); ^13^C-NMR (100 MHz, CDCl_3_): *δ* 175.95 (–COO–), 169.70 (–COO–), 150.68 (–CH=C), 109.77 (–CH=C), 82.04 (C–OH), 81.45, 65.86, 59.12, 56.68, 55.51, 51.31, 50.58, 50.13, 49.57, 47.09, 43.89, 42.53, 40.80, 38.48, 38.31, 37.97, 37.07, 34.34, 32.24, 30.70, 29.69, 28.20, 25.61, 23.93, 21.03, 19.47, 18.30, 16.73, 16.70, 16.30, 15.96, 14.78. benzene ring: 136.63, 128.62, 128.38, 128.19. m.p.: 143.0–145.9 °C. HR-MS (ESI) *m*/*z*: 673.4945 [M + H]^+^, calcd for: C_43_H_65_N_2_O_4_ 673.4866. 

### 4.3. Biology Evaluation

The human cervical cancer cell line (Hela), human hepatocellular carcinoma cell line (HepG-2), human gastric cancer cell line (BGC-823) and human neuroblastoma cell line (SY-SY5Y) were obtained from the Chinese Academy of Medical Sciences and Peking Union Medical College. Fetal bovine serum (FBS) and RPMI 1640 (DMEM) medium, penicillin and streptomycin were obtained from Thermo Technologies. 6-diamidino-2-phenylindole (DAPI) was obtained from Molecular Probes/Invitrogen Life Technologies (Carlsbad, CA, USA). The cultures of the cells were maintained in RPMI 1640 or Dulbecco’s Modified Eagle’s Medium (DMEM) supplemented with 1% (*v*/*v*) penicillin/streptomycin and 10% (*v*/*v*) fetal bovine serum under a humidified atmosphere containing 5% CO_2_ at 37 °C.

The stock solutions of **BA** derivatives were dissolved in dimethyl sulfoxide (DMSO; Sigma, St. Louis, MO, USA) and added at various concentrations to the cell culture. Cellular morphologies were observed using an inverted fluorescence microscope (Olympus IX71, Tokyo, Japan), a plate reader (BIORAD 550 spectrophotometer, Bio-Rad Life Science Development Ltd., Beijing, China), and a Canton 2 flow cytometer (BD, New York, NY, USA).

#### 4.3.1. Antitumor Activity

The antitumor activity of **BA** derivatives was evaluated on Hela, HepG-2, BGC-823, SY-SY5Y cell lines using the MTT assay. The density of all cells was 2 × 10^3^ cells/well plated in a 96-multiwell plate in RPMI 1640 or DMEM containing 10% FBS for 24 h at 37 °C with 5% CO_2_. Then, cells were treated for 48 h with the required concentrations (3.125, 6.25, 12.5, 25, 50, or 100 μM) of **BA** derivatives dissolved with the vehicle DMSO. Each plate contained control group, blank group and drug group. After that, 20 μL MTT in phosphate buffered saline (PBS, 5 mg/mL) was added to each well, and the plates were incubated at 37 °C for 4 h, then we removed the supernatant and adding dimethyl sulfoxide (DMSO, 150 μL) to dissolve the MTT formazan. The optical density (OD) for each well was measured on a BIORAD 550 spectrophotometer plate reader at a wavelength of 550 nm. The above tests were repeated three times in parallel. The proliferation inhibition rates of tumor cells were calculated by {1 - [OD_550_ (Drug group)/OD_550_ (Blank group)]/[OD_550_ (Control group) - OD_550_ (Blank group)]} × 100%. Compounds with concentration less than 25 μM and proliferation inhibition rates higher than 50% were rescreened. The concentrations of **BA** derivatives were required at 1.5625, 3.125, 6.25, 12.5, or 25 μM to calculated IC_50_ values for rescreened.

#### 4.3.2. Morphological Analysis

Hela cells in the logarithmic growth phase were plated onto 6-well plates at a density of 2 × 10^4^ cells/mL for 24 h at 37 °C in a humidified atmosphere with 5% CO_2_. Additionally, each group was treated with 5 μM **BA** and compound **7e** for 48 h. Cell culture medium was discarded, and the cells were washed twice with PBS. The cells were fixed with 400 μL 4% paraformaldehyde (pH = 7.4) for 10 min and then washed twice with PBS. Then fixed cells were stained with DAPI at the concentration of 1 mg/mL for 20 min in the dark, and cell morphological changes were observed using a fluorescent inverted phase-contrast microscope at a magnification of 100 ×.

#### 4.3.3. Apoptosis Analysis Using Annexin V-FITC/PI Staining

Hela cells in the logarithmic growth phase were plated onto 6-well plates at a density of 4 × 10^4^ cells/mL at 37 °C in a humidified atmosphere with 5% CO_2_. After incubation for 24 h, Cell culture medium was discarded, and cells were treated with various concentrations (0, 1, 2, or 4 μM) of compound **7e** for a further 48 h. Then, cells were collected, washed twice with cold PBS, and centrifuged at 2400 rpm for 10 min. The resulting pellet was mixed with 200 μL of binding buffer of the Annexin V-FITC kit; then, 5 μL of FITC-labeled annexin V was added and mixed gently. After incubation at 4 °C for 10 min in the dark, 5 μL of PI was added and mixed gently. Then, the cells were immediately analyzed with a flow cytometer at 488 nm [26,27].

### 4.4. Statistical Analysis

All results were expressed as means ± standard derivation (SD) of three independent experiments. The statistical analysis was performed by SPSS software (Version 20.0, International Business Machines Corp. New York, NY, USA) to analyze the variance. One-way analysis of variance (ANOVA) was performed to determine the significance between groups; *p* < 0.05 was considered to be statistically significant.

## 5. Conclusions

In this paper, a series of different **BA**-nitrogen heterocyclic derivatives were designed and synthesized. All of them were characterized by ^1^H-NMR, ^13^C-NMR (Appendix A) and were screened for cytotoxic activity employing a panel of four cell lines, including Hela, HepG-2, BGC-823 and SK-SY5Y cells, using the MTT assay. From these data analyzed with MTT, it was evident that almost all derivatives exhibited higher cytotoxicity for all tested cell lines compared to **BA**. Compound **7e** was found to be the most likely drug candidate, showing that IC_50_ values were 12-fold toxic in vitro than **BA**-cell Hela. As shown by DAPI and Annexin V-FITC/PI staining, it was found that compound **7e** mainly acted by inducing early apoptosis. Based on the above, compound **7e** showed bright prospects and is valued for further study.

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
