# Peer review of "Betulinic Acid-Nitrogen Heterocyclic Derivatives: Design, Synthesis, and Antitumor Evaluation in Vitro"

_molecules, 2020, doi:10.3390/molecules25040948_

Round 1
Reviewer 1 Report
The authors report the synthesis of a chemo library of antitumor candidates by hemisynthesis from betulinc acid. Products were evaluated for their cytotoxicities and the best candidate was further evaluated on morphological effect on Hell cells. Furthermore, the induced apoptosis was analyzed by flow cytometry.
The chemistry work deal with classical chemical transformations but the biological results are the biological results are relevant. The article is well written and informative for the introduction and the results sections, but the discussion is shallow. For this last reason I recommend a major revision.
The experimental section is well described, and NMR spectra are from very good quality.
Minor comments:
In vitro should be in italic font
In the introduction it is not necessary to make a statement about the NMR/HRMS characterizations of the products. This is mandatory for all publications.
In the discussion 79% not 79,3%
Major revision
The discussion section is not informative: it is a small conclusion with no relevant discussion. The change of solvent (THF to DCM) is not a major result.
Authors should provide a clear discussion on the structure / activity relationship. The effect of the heterocycles is barely discussed, and the importance of the alkalinity is just mentioned without explanation and the reader must find by himself the conclusion.
This section must rewrite with clear explanations and statements.
Author Response
Authors' Responses to Reviewer's Comments (Reviewer 1)
Author's Notes
We tried our best to improve the manuscript and made some changes in the manuscript. These changes will not influence the content and framework of the paper. Special thanks to you for your good comments.
Point 1: In vitro should be in italic font
Response 1: Thank you very much for your comments. We have re-read and revised the whole manuscript, and corrected the format of in vitro. Some examples are as follows:
" in vitro " was changed to "in vitro "
Point 2: In the introduction it is not necessary to make a statement about the NMR/HRMS characterizations of the products. This is mandatory for all publications
Response 2: In the abstract, " the NMR/HRMS characterizations of the products" was removed. Specific as follows:
Abstract: Betulinic acid (BA) is a star member of the pentacyclic triterpenoid family, which exhibits great prospects for anti-tumor drug development. In an attempt to develop novel anti-tumor candidates, 21 BA–nitrogen heterocyclic derivatives were synthetized, in addition to four intermediates, 23 of which were first reported. Moreover, they were screened for in vitro cytotoxicity against four tumor cell lines (Hela, HepG-2, BGC-823 and SK-SY5Y) by standard methylthiazol tetrazolium (MTT) assay. The majority of these derivatives showed much stronger cytotoxic activity than BA. Remarkably, the most potent compound 7e (half maximal inhibitory concentration (IC50) was 2.05±0.66 μM) was 12-fold more toxic in vitro than BA-treated Hela. Furthermore, multiple fluorescent staining techniques and flow cytometry collectively revealed that compound 7e could induce the early apoptosis of Hela cells. Structure-activity relationships were also briefly discussed. The present study highlighted the importance of introducing nitrogen heterocyclic rings into betulinic acid in the discovery and development of novel anti-tumor agents.
Point 3: In the discussion 79% not 79,3%
Response 3: In the discussion, " 79,3%" was changed to " 79% "
Point 4: The discussion section is not informative: it is a small conclusion with no relevant discussion. The change of solvent (THF to DCM) is not a major result.
Authors should provide a clear discussion on the structure / activity relationship. The effect of the heterocycles is barely discussed, and the importance of the alkalinity is just mentioned without explanation and the reader must find by himself the conclusion.
Response 4: Thank you very much for your comments. According to your advices, the section of discussion was rewritten as follows:
BA is widespread in natural plant and Chinese herb medicine, used for prevention and treatment tumor. Now, there are large number of betulinic acid derivatives that have been synthesized [21-24]. In this report, a series of different BA–nitrogen heterocyclic derivatives were designed and synthesized to improve biological activity and hydrophilicity. After introducing different nitrogen heterocycle in the 3-hydroxyl/28- carboxyl of BA using the ester condensation reaction, the majority of these derivatives showed much stronger cytotoxic activity than BA.
In chemical synthesis, introducing succinic anhydride in the C-3 of BA was explored, and the reaction solvent was changed from THF to DCM. This reaction was simple, mild and controllable, with a yield of 79%, which was suitable for the synthesis of such compounds in the future. In structure-activity relationship, we could easily find that the structural modification site of BA and linked with different nitrogen heterocyclic rings on BA had an effect on the antitumor activity of the BA derivatives in vitro. In general, as observation for compounds 7a, 7c, 7d, 7e and 5a-5d, structural modification at positions C-28 and C-3 could improve antitumor biological activity, especially structural transformation of C-28 might have more potential to enhance cytotoxicity on the same series of tumor cells; Besides, different nitrogen heterocyclic rings on BA also influenced their activity (compound 11e > compound 11a, 11b, 11c, 11d), and the alkalinities of the different nitrogen heterocyclic rings were positively correlated with their activities, which might be likely associated with increasing bioavailability and altering an extracellular weak acidic microenvironment with further verification [25].
Reviewer 2 Report
The authors present their work on synthesis and in vitro anti-tumor evaluation of betulinic acid-nitrogen heterocyclic derivatives.
Considering the applications and properties of Betulinic Acid and its derivatives, it is particularly interesting to develop new derivatives.
However, some clarifications are needed before the capability to publish this article.
Why compounds 4a-d, 10 and 11a-d were not subjected to the cytotoxicity tests (Table 2) since only compound 11e was tested. It seems necessary to me to test these 9 compounds before publishing this article or to explain at least what was the point of synthesizing them especially concerning compounds 11a-d.
Many imperfections are to be noted in the manuscript.
In abstract line 16: only 21 BA-nityrogen heterocyclic derivatives were synthetized.
In all the manuscript, the word "compound" are not to be in bold.
In the chemical synthesis paragraph lines 63 to 69 the text is confused.
replace the text with :
"The syntheses of 21 BA–nitrogen heterocyclic derivatives were shown in scheme 1. BA was treated with potassium carbonate solution and benzyl bromide/1, 2-dibromoethane in DMF at 85 °C for 4h to obtain compound 2 and 6. Then compound 2 was treated with succinic anhydride and chloroacetic acid in DCM at 80 °C for 5 h catalyzed by EDCI/DMAP, and compound 3 and 10 were obtained. By further substitution with nitrogen heterocyclic ring (R) or reduction reaction, we got the compounds 4a-4d, 5a-5d and 11a-11e (Table 1). Compounds 7a, 7c-7e were obtained from compound 6 by substitution with nitrogen heterocyclic ring (R), compounds 9a-9d (Table 1) were obtained by oxidation and substitution reaction based on compound 6."
because : there is only 21 BA–nitrogen heterocyclic derivatives and not 25 and explanations about compounds are confusing.
In diagram 1: the group R corresponding to the derivatives e is missing.
It is mentioned 11a-11d which is to be replaced by 11a-11e
similarly for compound 7 replace 7a-7d with 7a, 7c-7e.
Figure 2 is not very legible.
In conclusion, the authors mentioned "All these compounds were screened", in this case where are the results, because only 16 compound and BA were tested.
The article must be completed by the authors in terms of results before it can be accepted.
Reviewer 3 Report
The manuscript describes synthesis of 25 derivatives of Betulinic acid by introducing various substituents on hydroxyl and/or carboxyl group. Presented chemistry is rather trivial but obtained analogs are well characterized. A great variety of substituents was introduced but no rationalization is given why these and no other groups were chosen. In vitro cytotoxicities of the obtained compounds against four cancer cell lines were tested. In general activity was high and often greater than for Betulinic acid itself. One compound particularly active against all four cancer cell lines was selected for further biological screening which showed that this compound could induce the early apoptosis of Hela cells. However, there is no structure-activity relationship discussion.
I would recommend this manuscript for publication in Molecules however major revision is necessary:
Rationalization for the introduced groups should be given. Structure-activity relationship discussion should be added.Other remarks:
line 31, should be “…drugs are developed…” line 40, why hydroxyl group in carboxyl moiety is called “primary hydroxyl group”? line 49, sentence starting “Barbara Eigenerova…” is completely not clear and should be rewritten. Figure 1, should be “…structure…” not “structures”. line 63, should be “…are shown in Scheme 1.” line 69, should be …reaction, starting from compound 6.” line 91, should be “It was further verified that small nitrogen heterocycles…”. point 4.2 Chemical Syntheses – Instead of “Compound 2” “Compound 3”, etc, full names of the obtained compounds should be given.Author Response
Please see the attachment

Round 2
Reviewer 2 Report
This corrected article can be published in present form
Reviewer 3 Report
I recommend this manuscript for publication in Molecules ater minor changes:
line 17, should be "...23 of which were reported for the first time." line 48, sentence startig from "Eignerova" is still unclear. The authors changed only the order of the first and last name what was unnecessary. This sentence should be rewritten in clear English.